

# Feathery and network-like filamentous textures as indicators for the crystallization of quartz from a silica gel precursor at the Rusey Fault, Cornwall, UK

Tim I. Yilmaz[1], Florian Duschl[2], Danilo Di Genova[3]

[1]Technical University Munich, Tectonics and Material Fabrics Section, Arcisstr. 21, 80333 Munich, Germany
[2]Geoscience Center of the University of Göttingen, Goldschmidtstr. 3, 37077 Göttingen, Germany
[3]LMU Munich, Department for Earth and Environmental Sciences, Theresienstr. 41, 80333 Munich, Germany
*Correspondence to*: Tim I. Yilmaz (tim.yilmaz@tum.de)

**Abstract.** Quartz crystals from a hydrothermal shear-zone-hosted quartz deposit (Rusey Fault, Cornwall, UK) show feathery textures and network-like filamentous textures. Optical hot-cathodoluminescence (CL) analysis and LA-ICP-MS investigations on quartz samples revealed that positions exhibiting feathery textures (violet luminescence) incorporate higher

amounts of Ca, As, Na, Mg, and K than quartz positions without feathery textures (blue luminescence). Raman spectroscopy investigations revealed the presence of a weak peak ('shoulder') at 507–509 $cm^{-1}$ in quartz affected by feathery textures, which we attribute to the presence of moganite, a microcrystalline silica variety.

The combined occurrence of feathery textures and network-like filamentous textures in quartz samples from the Rusey fault zone points to the presence of a silica gel precursor before or during the crystallization.

**Keywords.** Quartz, feathery textures, gel, moganite, hydrothermal quartz, cathodoluminescence, LA-ICP-MS, Raman

## 1 Introduction

The E–W-trending Rusey fault is ~100 km long and situated in the Culm basin of Cornwall, (UK) (Fig. 1). The studied outcrop is located on the northern coast of Cornwall in metasediments of the Variscan foreland (Fig. 2) at the contact

between the Crackington and Boscastle Formations of the Culm basin (Shackleton et al., 1982; Isaac and Thomas, 1998). The Crackington Formation is made up by cycles of fine sediments (sandstones, siltstones and mudstones) interpreted as parts of the lower Bouma sequences. The Boscastle Formation is made up by grey to dark grey and black slates with a moderately strong, gently dipping cleavage that has been interpreted as parts of the upper Bouma sequences (Thompson and Cosgrove, 1996; Isaac and Thomas, 1998). The Boscastle Formations stratigraphic position is Dinantian to Lower Namurian

to the south of the Culm basin, whereas the Crackington Formations is Namurian to Westphalian (Isaac and Thomas, 1998). Both formations have been considered as lateral counterparts of the Namurian in the Culm basin, in which the Boscastle Formation is supposed to be the more distal deposit (Thompson and Cosgrove, 1996). The British Geological Survey 1:50.000 geological map (British Geological Survey, 2013) (Fig. 2) as well as Thompson and Cosgrove (1996) suggest that



further fault-bounded lithological units of transitional nature between the Crackington Formation and the Boscastle Formation do exists at the studied outcrop on the coast.

Within the Rusey fault, hydrothermal quartz precipitated which shows (i) feathery textures and (ii) network-like filamentous textures. These microstructures occur in quartz coatings of so called cockade-like quartz coatings (Frenzel and Woodcock, 2014). Feathery textures generally appear in blocky to subhedral quartz grains (Gebre-Mariam et al., 1993; Moncada et al., 2012; Henry et al., 2014), while network-like filamentous textures in general occur in microcrystalline chalcedony (Duhig et al., 1992; Grenne and Slack, 2003; Little et al., 2004). Both, quartz and chalcedony precipitated under hydrothermal conditions. Feathery textures, microstructures commonly occurring in many hydrothermal quartz deposits, were first reported in quartz veins in Kingman, Arizona (Adams, 1920). Two models have been proposed to explain the origin of these textures: (i) epitaxial overgrowth of small quartz crystals on large existing quartz crystals (Dong et al., 1995) and (ii) crystallization from former fibrous, water-rich chalcedony (Sander and Black, 1988). Recently, Marinova et al. (2014) has reported that feathery textures are generally accepted as being a re-crystallization product from chalcedony in the context of having a gel precursor.

The feathery textures in our samples are frequently arranged on the intragranular growth zoning of both, anhedral to subhedral quartz grains and locally comb-shaped crystals (Fig. 3a,b). They are characterized by 5–20-µm-sized subgrains, which appear as splintery or feathery patterns under a standard petrological microscope with crossed polarizers due to slight optical differences in maximum extinction positions (Fig. 4). Subgrains of the feathery textures are arranged along the *c*-axes in a sub-parallel arrangement to each other forming filamentous bundles; the subgrain long-axes orientation within these bundles being <90° from the *c*-axis of the core. Locally they are restricted to growth zones and are accompanied by a high amount of fluid inclusions (Fig. 4).

Feathery textures are also accompanied by fluid inclusions (FI) that can be grouped in two types: (a) primary and/or pseudosecondary aqueous biphase FI within feathery textures (intragranular) (Figs. 3a, 4), and (b) secondary aqueous biphase FI that trace healed microfractures (intergranular). These microfractures affect both, feathery textures and otherwise FI-free crystal cores. Type-(a) inclusions occur along boundaries between subgrains or filamentous bundles. The shape of type-(a) inclusions is usually irregular or tubular-elongated with acicular ends, many of them are interconnected by minute channel-like features; type-(b) inclusions typically show irregular shapes. The size of fluid inclusions range from <5 µm to 10 µm.

Despite the recent advantages in studying these textures our understanding of the origin of this texture is still incomplete mostly because few data have been published. Here we report the obtained results from a multidisciplinary approach based on optical hot-cathodoluminescence (CL) analysis, LA-ICP-MS and Raman spectroscopy investigations on quartz crystals in order to obtain chemical insight into these microstructures.



## 1.2 Results and Discussion

### 1.2.1 Cathodoluminescence (CL)

They are characterized by 5–20-µm-sized subgrains, which appear as splintery or feathery patterns under a standard petrological microscope with crossed polarizers due to slight optical differences in maximum extinction positions (Fig. 4).

Subgrains of the feathery textures are arranged along the $c$-axes in a sub-parallel arrangement to each other forming filamentous bundles; the subgrain long-axes orientation within these bundles being <90° from the $c$-axis of the core. Locally they are restricted to growth zones and are accompanied by a high amount of fluid inclusions (Fig. 4).

Feathery textures are also accompanied by fluid inclusions (FI) that can be grouped in two types: (a) primary and/or pseudosecondary aqueous biphase FI within feathery textures (intragranular) (Figs. 3a, 4), and (b) secondary aqueous

biphase FI that trace healed microfractures (intergranular). These microfractures affect both, feathery textures and otherwise FI-free crystal cores. Type-(a) inclusions occur along boundaries between subgrains or filamentous bundles. The shape of type-(a) inclusions is usually irregular or tubular-elongated with acicular ends, many of them are interconnected by minute channel-like features; type-(b) inclusions typically show irregular shapes. The size of fluid inclusions range from <5 µm to 10 µm.

Despite the recent advantages in studying these textures our understanding of the origin of this texture is still incomplete mostly because few data have been published. Here we report the obtained results from a multidisciplinary approach based on optical hot-cathodoluminescence (CL) analysis, LA-ICP-MS and Raman spectroscopy investigations on quartz crystals in order to obtain chemical insight into these microstructures.

### 1.2.2 LA–ICP–MS

A laser ablation line within one thin section (Fig. 6) was defined to examine the chemistry of the grains of the quartz coatings including locally well-developed feathery textures. As reported by several studies (Rusk et al, 2008; Flem and Müller, 2012; Rusk, 2012), the main substituent for $Si^{4+}$ in quartz is $Al^{3+}$ with $Li^+$ or $Na^+$ balancing the missing positive charge in the crystal lattice. Moreover, various other trace elements such as B, Ge, Fe, H, K, Na, P and Ti tend to be incorporated as lattice-bound impurities. Sb may also play a role, particularly in hydrothermal quartz (Rusk et al., 2011).

Other commonly occurring elements including Ca, Cr, Cu, Mg, Mn, Pb, Rb, and U are suggested to be input from fluid or solid inclusions, which may occasionally influence mass spectrometric analysis (Müller et al., 2003; Flem and Müller, 2012). Within the laser ablation line shown in Fig. 6, the Si content remained constant throughout the measured part of the thin section. The intensities of Sb vary strongly but essentially remain stable over the entire line. Elements such as Ca, As, Na, Mg, and K show a significant increase in quartz grains with feathery textures. Furthermore, the peaks of these elements

correlate locally. At some positions, the intensities increase dramatically, although these variations are probably related to solid inclusions.



The increase in Mg as well as Na is exceptionally high and therefore cannot be influenced only by fluid inclusions (Müller et al., 2003). In addition, only small volumes of fluid inclusions are visible in the thin section. Furthermore, a decrease in Si content should be expected where fluid inclusions affect the ablation current (Müller et al. 2003); this result was not observed. Other important elements such as Ti, Li, and B are below the detection limit and were therefore not considered in the results.

### 1.2.3 Raman spectroscopy

Raman spectroscopy is a spectroscopy technique, which offers several advantages over other spectroscopic and analytical methods as little or no sample preparation is required. The technique has a non-destructive character and it allows analysis of extremely small sample volume with a sub-millimeter resolution (in the order of microns).

As reported above, the origin of feathery textures may be explained with the re-crystallization of former fibrous, water-rich, chalcedony. Chalcedony is microcrystalline silica composed of nano-scale intergrowths of α-quartz and moganite (Heaney, 1993). Within chalcedony and other microcrystalline silica, varieties between 5 wt% and 20 wt% of moganite may crystallize (Heaney and Post, 1992).

Raman spectroscopy investigations performed on $SiO_2$ samples from hydrothermal deposits, cherts, and flints have demonstrated that moganite in microcrystalline quartz/chalcedony can be detected (Kingma and Hemley, 1994; Hopkinson et al., 1999; Götze et al., 1998, 1999; Rodgers and Cressey, 2001; Rodgers and Hampton 2003; Pop et al., 2004; Rodgers et al., 2004; Heaney et al., 2007; Schmidt et al., 2012; Sitarz et al., 2014). Results shown that the main Raman scattering bands are located at 462–465 $cm^{-1}$ and 501–505 $cm^{-1}$ for α-quartz and moganite respectively.

A thin section from the quartz zone of the Rusey fault, specifically the quartz coatings surrounding wall rock and gouge fragments, was analyzed in order to examine the origin of the feathery textures. Over the entire thin section, 40 measurements were conducted in the quartz grains exhibiting feathery textures. Three representative measurements are shown in Figs. 8a,b and 9a,b. The main quartz band shifted between 462 and 464.5 $cm^{-1}$. Secondary bands were detected at ~124–127, ~200–203, 259–263, 351–354, 396–399, 694, 805–807 $cm^{-1}$. In addition, a weak shoulder of the main band was detected between 507 and 509 $cm^{-1}$ (Figs. 8b and 9a). This weak contribution was found within feathery textures but not within the crystal cores (Fig. 8a). This weak shoulder was also observed in the measurements of a pure quartz sample (Fig. 9b).

As we mentioned above, moganite occurs with α-quartz in chalcedony. A comparison of our results obtained from Rusey samples with those obtained from pure quartz crystals revealed that the band at ~503 $cm^{-1}$ appears also in α-quartz, in accordance with Pop et al. (2004). Within α-quartz, these peaks might indicate the substitution of Si by Al. Additionally the weak peaks at 507 and 509 $cm^{-1}$ measured within the Rusey samples may indicate moganite within the feathery textures.

Network-like filamentous textures are exhibited locally in quartz of the quartz coatings surrounding wall rock fragments (Fig. 10a–c). They are localized within a zone confined by a smooth irregular outline (Fig. 10a) and are characterized by fine



(<5 μm) irregular-shaped pigment inclusions probably composed of Fe-oxides (Fig. 10c). The pigment inclusions are not restricted by grain boundaries and occur within quartz exhibiting and not exhibiting feathery textures. These textures are similar to those described by Duhig et al. (1992) in chalcedony.

The similarity to gel polymerization textures (Brinker and Scherer, 1985; Shih et al., 1989; Scherer, 1999) might indicate the relics of a polymerized material before overgrowth by blocky to subhedral and partly euhedral hydrothermal quartz. These relic polymerization textures might be an evidence of a silica gel precursor but further investigations need to be conducted.

### 1.3 Conclusions

Laser ablation measurements indicate that the incorporation of elements such as Ca, Mg, As, Na, and K within feathery textures may exhibit red to purple CL colors and are most likely caused by point defects (Götze et al., 2009). The increased incorporation of various elements might be related to increased growth rates (Müller, 2000) controlled by changes in temperature or pressure (Ramseyer et al., 1988; Götze 1996) or pH conditions (Ramseyer and Mullis, 1990; Perny et al., 1992).

An increased growth rate leads to an increase in element intake (Müller, 2000), indicating that feathery textures have precipitated faster than their blue CL cores. The Ti-values, which are below the detection limit, indicate precipitation temperatures below 400 °C (Götte and Ramseyer, 2012). Those and other values such as those of B and Li, which also fell below the detection threshold, indicate a crustal origin of formation fluids (Götte and Ramseyer, 2012). Furthermore, yellow luminescence emission bands, which correlate with yellow CL in the Rusey samples, indicate high concentrations of lattice defects probably generated by the rapid crystallization of a non-crystalline precursor (Götze et al., 1999).

Raman spectroscopy revealed the occurrence of a weak peak at 507–509 cm$^{-1}$ (Figs. 8b and 9a), which we attribute to the presence of moganite within the feathery textures. The presence of moganite could be a direct link to chalcedony within the feathery textures (Heaney, 1993).

The presence of locally occurring network-like filamentous textures (Fig. 8a–c), which have appearances similar to polymerization structures (Scherer, 1999), may indicate a polymerization stage of a possible silica gel phase (Duhig et al., 1992).

In conclusion, the presence of feathery textures and network-like textures textures may indicate the crystallization of a chalcedony precursor (Sander and Black, 1988; Duhig et al., 1992; Marinova et al., 2014), thus indicating the result of re-crystallization from a silica gel derived metastable $SiO_2$ phase (e.g. amorphous silica) (Oehler, 1976).

### Acknowledgments

We are grateful to Stephen F. Cox for his helpful discussions and Alfons van den Kerkhof for his discussions and support during CL measurements and interpretation. We also thank Ottomar Krentz for field work support, Kai-Uwe Hess for Raman



measurements and interpretation, and Klaus Mayer for sample cutting and preparation. This study was financially supported by the German Academic Exchange Service (DAAD) within the Australia–Germany Joint Research Cooperation Scheme (project 56267246). Tim Ibrahim Yilmaz gratefully acknowledges financial support by the Leonhard Lorenz Foundation (grant 826/12) and the TUM Graduate School (TUM GS).

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





**Figure 1: Geological sketch map of Cornwall showing Devonian to pre-Devonian mica and amphibolite schists, Devonian to Carboniferous basins (indicated by short dashed lines) and Upper Carboniferous to Lower Permian granitic intrusions. Furthermore two structures, the Rusey fault (RF) and the Sticklepath-Lustleigh fault (SP) are indicated. The dashed rectangle indicates the position of Fig. 2. Modified after Leveridge and Hartley (2006).**





**Figure 2: Geological and structural map of the Rusey fault zone area showing the Boscastle Formation to the south and Crackington Formation to the north of the Rusey fault zone. Most faults have an NW-SE to WNW-ESE trend only a few faults have a NE-SW to ENE-WSW orientation. The position of the Rusey fault zone is indicated by the black arrow. Modified after British Geological Survey 1: 50.000 geological map (British Geological Survey, 2013).**





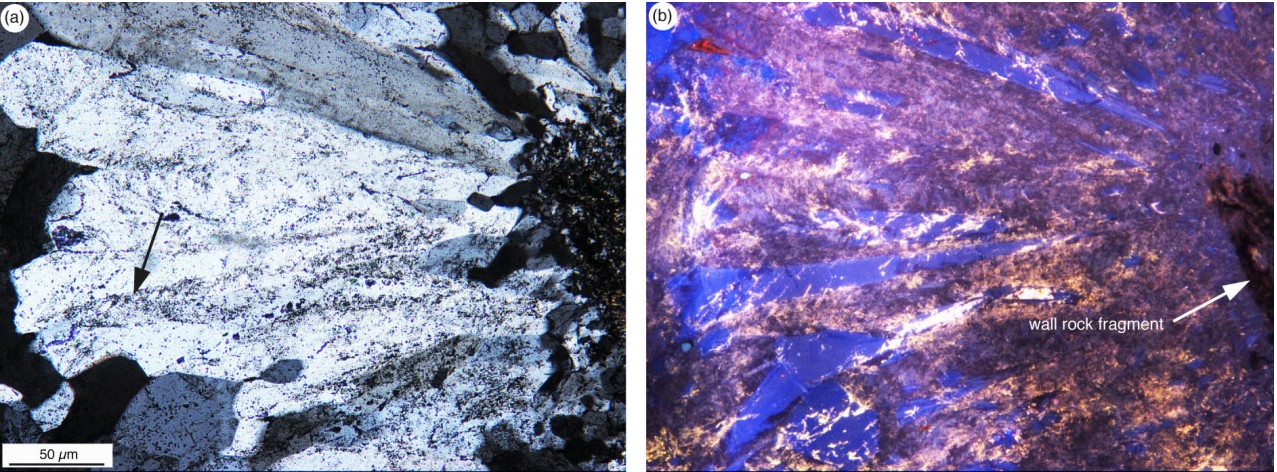

**Figure 3: Photomicrographs of a quartz coating surrounding a wall rock fragment from the Rusey quartz zone; (a) X pol; (b) CL. (a) A wall rock fragment is surrounded by quartz, which increases in size from the fragment toward the left. The large comb quartz shows zones with densely distributed fluid inclusions (black arrow). (b) The CL image reveals that the comb quartz is made up by a core with partly euhedral faces. That core shows initial blue luminescence colors, a patchy area with violet luminescence is representing feathery textures, and yellowCL represents fluid inclusion trails.**

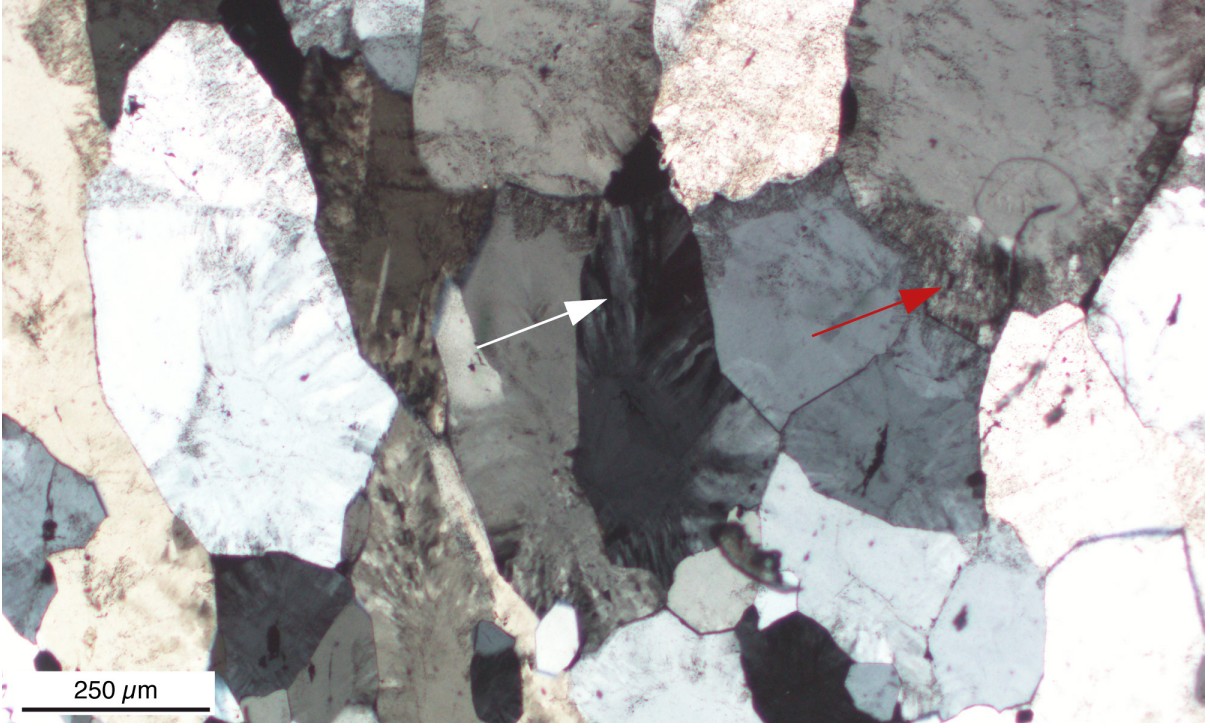

**Figure 4: Photomicrograph of blocky anhedral to subhedral quartz crystals of the quartz coating from the quartz zone (Rusey fault) showing feathery textures as indicated by the white arrow. The red arrow indicates feathery textures appearing within a zone in which a high amount of fluid inclusions is situated (non-oriented sample TY33X4; X pol).**



**Figure 5: Photomicrographs (a and c) and CL images (b and d) of anhedral to subhedral quartz grains exhibiting feathery textures. (a) Splintery or fibrous appearance of feathery textures, as indicated by the white arrow. The outer dashed line indicates the grain boundaries of one quartz grain, and the inner dashed line indicates a core with no feathery textures; X pol. (b) CL in which the patchy area of the feathery textures is shown by purple to reddish-brown colored with bright blue patches; the core is show by intense blue luminescence. The patchy violet area makes up ~70–80% of the image. (c) Photomicrograph of subhedral quartz grains locally showing feathery textures. The white dashed lines indicate intergranular zoning features (revealed in CL mode within (d)) X pol. (d) CL reveals red zoning features, which are indicated by black arrows. White arrows indicate growth zoning features within blue cores appearing in various shades of blue. The patchy violet area representing positions of feathery textures makes up ~60–75% of the image (non-oriented sample TY39X2).**





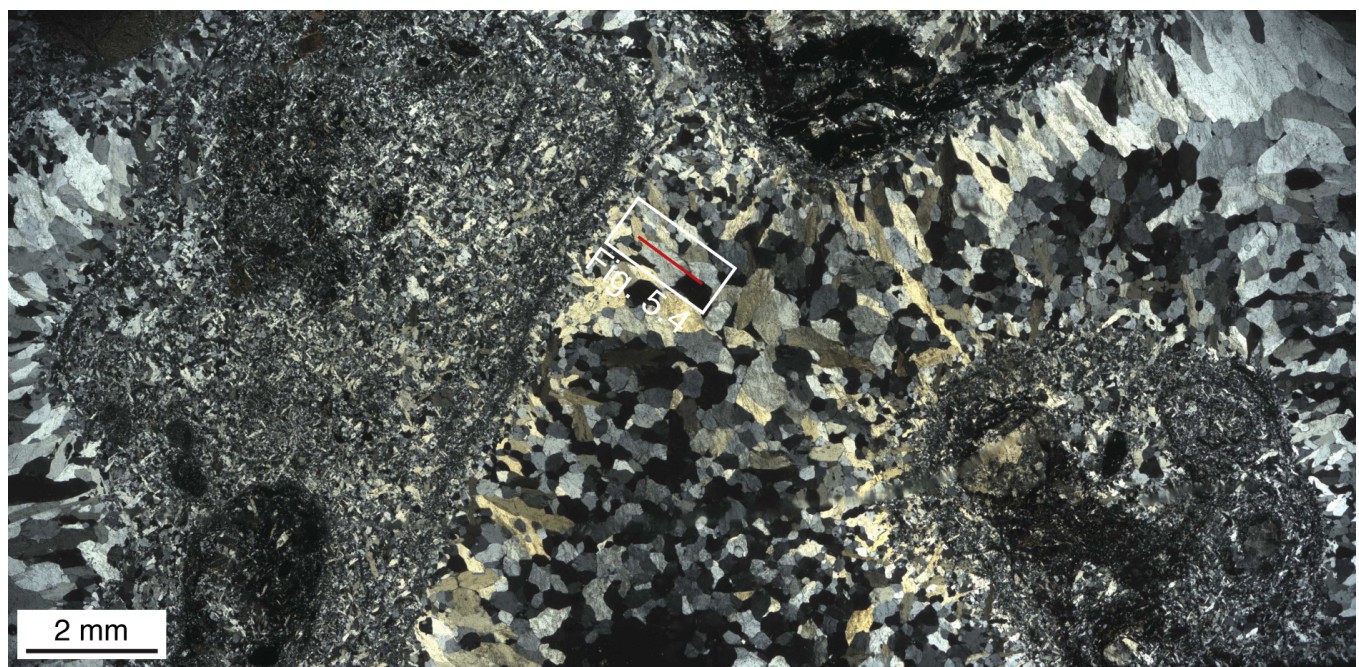

**Figure 6: Thin-section scan of sub-rounded and quartz-coated gouge fragments from the quartz zone the Rusey fault. The red line shows the analyzed LA–ICP–MS line. The white rectangle indicates the position represented in the lower part of Fig. 7, in which quartz grains with and without feathery textures have been investigated. Oriented sample 5198B1, X pol.**





**Figure 7: Laser ablation of quartz with and without feathery textures indicated by the red line within the photomicrograph and its corresponding spectra. The laser ablation line is cross-cutting three quartz grains. The bright quartz grain on the left does not exhibit feathery textures. The large central grain contains feathery textures but only at the indicated positions (feathery textures/no feathery textures). This grain contains a euhedral core, which is not visible in this extinction position. The quartz grain to the right is completely made up by feathery quartz, which is not visible in this extinction position. Element spectra of Si, K, Na, Al, Mg, As, Sb, Fe, and Ca shown are indicated in the legend below the photomicrograph. From left to right, the intensities of Ca, As, Mg, K, Al, and Na within the feathery texture increase. The dark line above the red line represents an earlier laser ablation line, which is not considered in this chapter. Oriented sample 5198B1, X pol.**

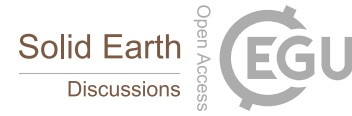



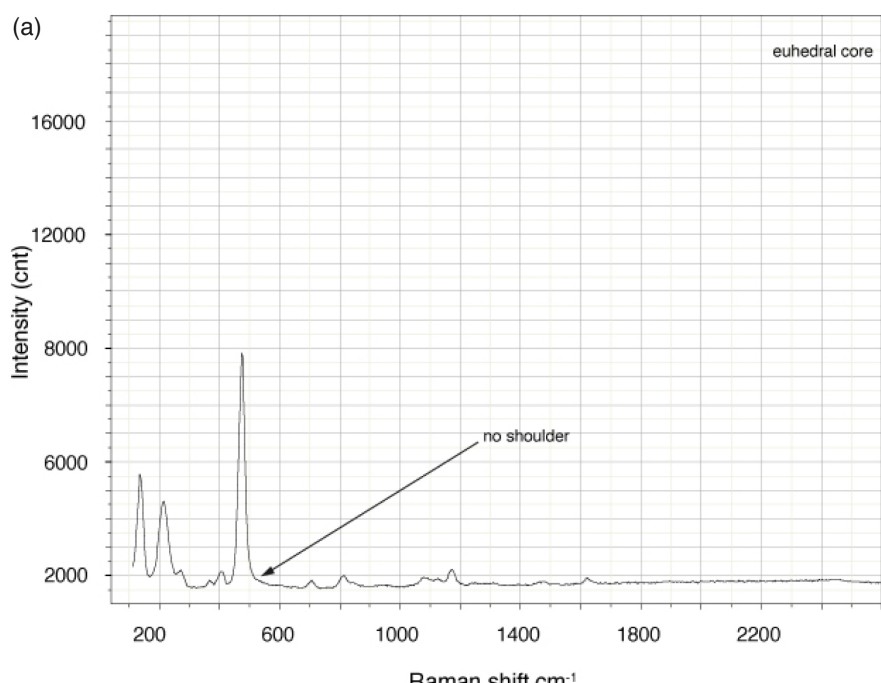

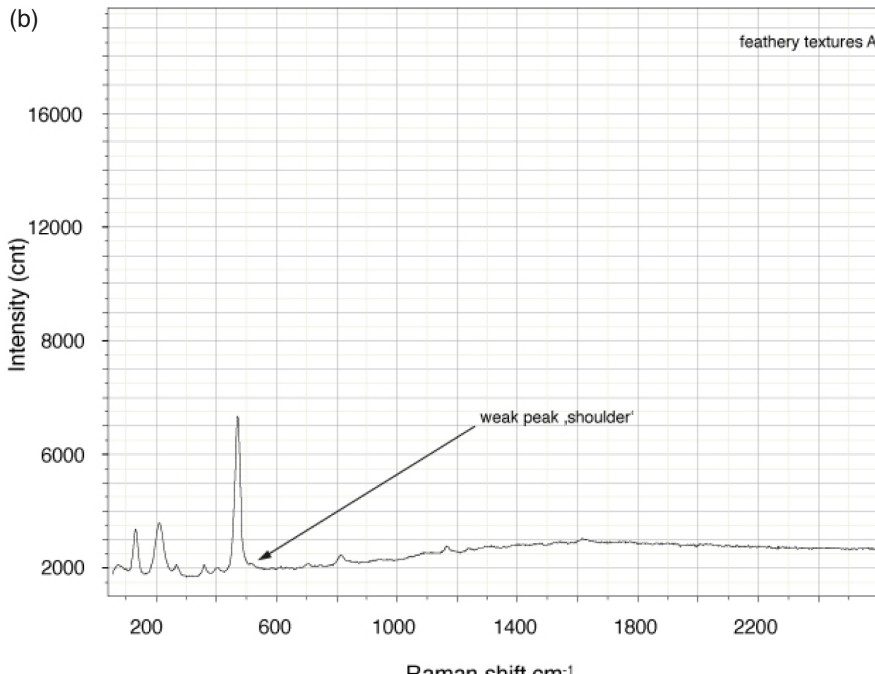

**Figure 8: Raman bands of SiO$_2$ samples. (a) Raman spectrum of a euhedral core of a quartz grain from a quartz coating in the Rusey fault zone showing no shoulder. (b) Raman spectrum from feathery textures from a quartz coating in the Rusey fault zone showing a shoulder at ~507 cm$^{-1}$.**

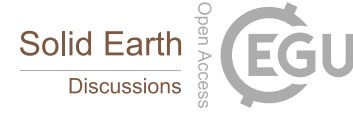

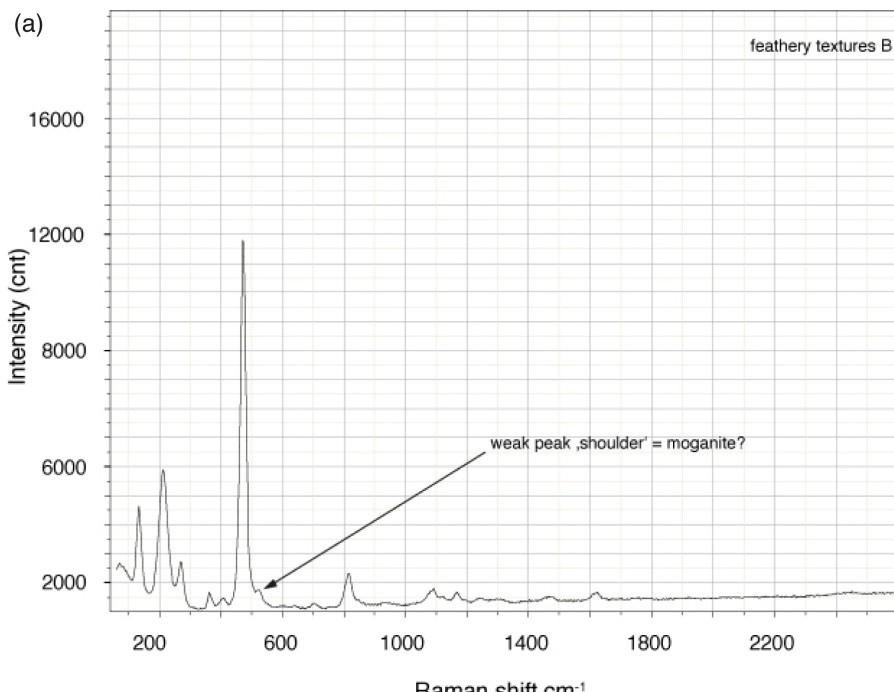

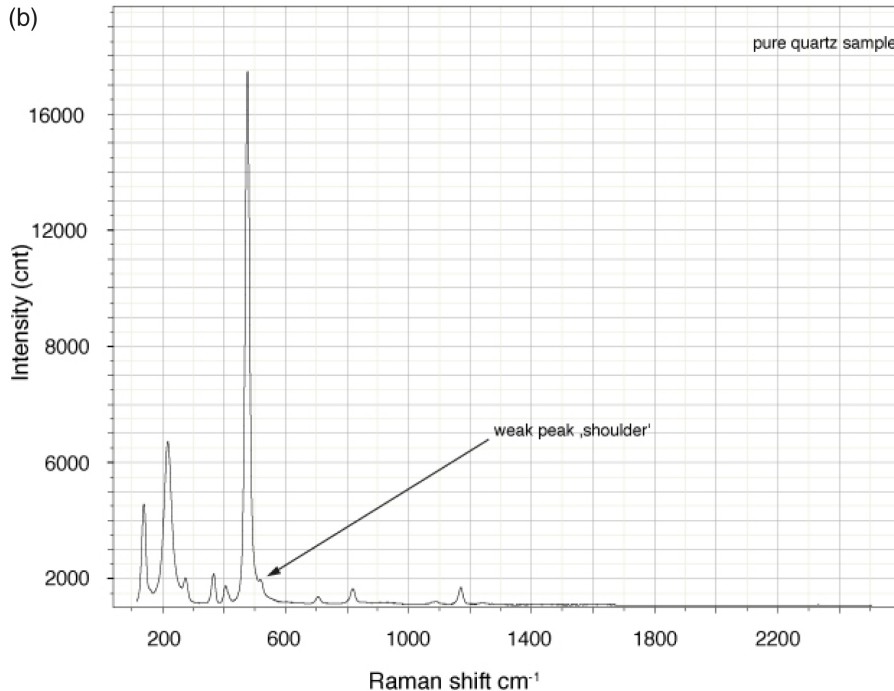

**Figure 9: (a) Raman spectrum obtained in feathery textures from the quartz coatings at the Rusey fault zone showing a shoulder at ~509 cm[-1]. (b) Raman spectrum from a α-quartz sample from the LMU archive shows a shoulder at ~503 cm[-1].**



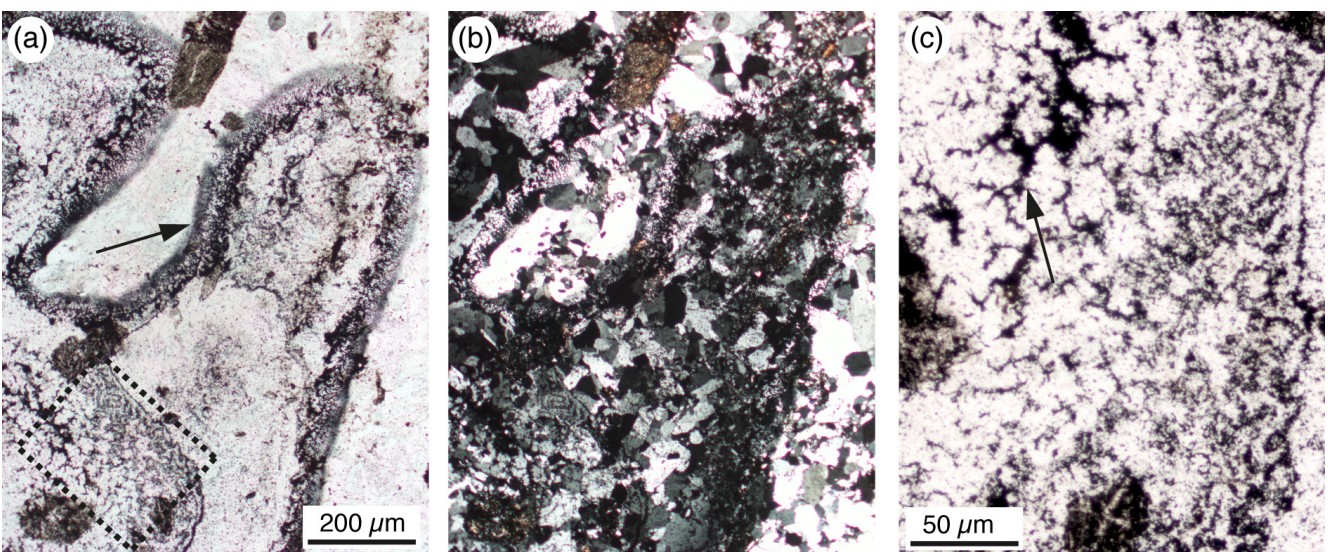

**Figure 10: Photomicrographs of blocky anhedral to subhedral quartz from the quartz coating in the Rusey samples. (a) Network-like filamentous and/or dendritic textures are made up by dark pigments most possibly µm-sized Fe-oxides, which are restricted by a irregular and diffuse boundary (indicated by the black arrow); the rectangle indicates the position of (c); II pol. (b) Quartz is made up by blocky anhedral to subhedral quartz; X pol. (c) A zoom into the dense distribution of the particles reveals that the inclusions form a network-like or dendritic texture (black arrow); II pol. Oriented sample TY31C1.**