# Peer review of "Feathery and network-like filamentous textures as indicators for the crystallization of quartz from a silica gel precursor at the Rusey Fault, Cornwall, UK"

_Solid Earth, 2016_

## Author Comment (AC1) · 5 Apr 2016

[revised manuscript text omitted]

**1.2 Results and Discussion**

**1.2.1 Cathodoluminescence (CL)**

Optical cathodoluminescence (CL) technique can be used to detect quartz and to reveal the processes of crystal growth, deformation, recrystallization, and alteration (Götze et al., 2001). This method can reveal zoning features within (hydrothermal) quartz crystals and can help to identify various quartz generations (Ramseyer et al., 1988; Ramseyer and Mullis, 1990).

The CL of quartz crystals from coatings exhibiting feathery textures is generally characterized by an intense blue (initial blue) core (~20–40%), which is surrounded by a purple to weak red to reddish-brown patchy area. The blue core of the quartz grain shows locally euhedral faces (Fig. 3b) whereas the patchy area represents feathery textures made up by quartz fibers (Fig. 3a,b). Note that CL colors strongly depend on the duration and intensity of electron radiation (Ramseyer et al., 1988). In particular, within hydrothermal quartz, short-lived blue luminescence may disappear after several seconds of radiation (Fig. 5a–d). The estimated percentage obtained by image analysis (ImageJ) of quartz exhibiting feathery textures is ~60 to 85% (Fig. 5b,d).

The patchy appearance of the feathery textures represents observed filamentous bundles under crossed polarized light. Grains with feathery textures locally show blue cores of hydrothermal quartz, which are locally highlighted by oscillatory growth zonings. These growth zonings appear in various shades of blue (Fig. 5d). Furthermore, red zoning features were observed, which are located within the feathery textures and in the blue cores (Fig. 5d). The zoning features are intragranular and demonstrate the primary growth character of both the blue cores and the irregular and patchy reddish seams. The quartz grains of the coating are traversed by fine irregular networks exhibiting bright yellow CL (Fig. 5b,d).

[revised manuscript text omitted]

---

## Referee Comment (RC1) · Anonymous Referee #1 · 16 May 2016

Dear Editor and Authors,

The manuscript deals with the formation of hydrothermal quartz in veins, an argument on which experimental data and observations are certainly scarce. This makes the article worth to be considered for a large audience.

I've found the manuscript well written, with a detailed description, rich of data and observations. The focus on a enrichment of Ca, Mg, Na, As, and K in the feathery structures is correct, representing a new finding, thus contributing to the understanding of the described microstructures.

[Figure]

The presented data support the proposed interpretation; the figures are all necessary. However, Fig.1 and 2 can be reasonably summarized in one, only.

I suggest to better address the "geological framework" to the described theme, giving more information on the geological conditions which favored the hydrothermal circulation along the RF fault zone. From the map given in Fig. 1 and 2, I've noted that post-hercynian plutons are indicated to the S and SE of the study area. Is the activity of the RF fault coeval with the cooling and exhumation of these plutons? Could it be possible to have an estimation on the age of the study quartz vein?

Finally, I've found this manuscript suitable to be published after implementing the geological setting (minor revisions).

---

## Referee Comment (RC2) · B. Rusk (Referee) · 20 Jun 2016

I've completed my review of "Feathery and network-like filamentous textures as indicators for the crystallization of quartz from a silica gel precursor at the Rusey Fault, Cornwall, UK" by Yilmaz et al. This manuscript presents CL images, LA-ICP-MS data, and Raman spectroscopic data of quartz infill from a vein in the UK. The main conclusion of the manuscript is that the feathery textures observed both optically and in CL are direct evidence for recrystallization of chalcedony to quartz. I have to admit that I have never read this journal and I am not familiar with its typical content. I couldn't

find any clear indications to authors or reviewers about manuscript length or style I apologize if I missed it. I therefore have a difficult time as to making a recommendation as to whether the article is suitable for publication or not. I think there are a number of problems and shortcomings with this manuscript that would prevent it from being accepted for many international journals; however, I do find the subject matter interesting. Even so, I don't find the conclusions convincing and I don't believe that the data presented in the manuscript actually has any direct reflection on the main conclusion of the manuscript-that is that feathery quartz necessarily is derived from recrystallization of chalcedony,

Below I give my major concerns with the manuscript.

In terms of the style, the manuscript is very short, and the style more typical of an abstract or a proceedings from a conference, than for journal publication. The introduction is the longest section of the entire paper and rather than present the framework or context of the science presented, it is a geologic background combined with a first telling of the results of the study and even contains the ultimate conclusion of the study-that feathery quartz results from recrystallization of chalcedony. The paper has no separate results and discussion section, which is not typical of a scientific paper. Instead the results and interpretations are mixed together for each of the 3 analytical techniques, and there is no section that integrates and synthesizes all of the data together to advance our scientific understanding. There is no description of the analytical methods used in the study, which is also atypical. Lastly, and most surprising with regards to the style of the manuscript, is that the introduction and the first section of the results section are a word for word copy and paste of 2 entire paragraphs. There is no need to be repetitive in scientific writing, and certainly copying and pasting text in multiple sections of a paper is not good presentation.

I terms of the results, the key CL textures identified and continuously refereed to are the feathery textures and the "network-like filamentous textures". No example of CL images of the "network like filamentous textures" are given, so it is not clear what

this description even means. Are these really the same texture with just two different names? Clarify. If they are the same, pick one and use it. Otherwise, the CL images aren't that different from the transmitted light textures. It's not clear whether there is a consistent and direct correlation between the optical feathery texture and the CL feathery texture or whether these are two discrete textures. Clarification here would improve our understanding. . ...

There are a number of problems with the LA-ICP-MS data. My suggestion is that the vast majority of the data represents the accidental ablation of micro and nano mineral and fluid inclusions. Where there are no inclusions, quartz is a clear mineral. All of the transmitted light images show abundant dark and cloudy spots that are inclusions of some kind. All of these inclusions, when ablated, will contribute to the LAICPMS signal. Feathery textures have more such inclusions than euhedral quartz, which is why elemental values go up in the feathery quartz. The elements you present results for, you admit yourself ,are typically caused by inclusions, and are not commonly present in quartz as demonstrated by many other studies. The results of a number of studies agree with this statement. These elements are not abundant in quartz; they are more common in fluid and mineral inclusions. Your elemental values are far higher than is reasonable to be structurally incorporated into quartz alone (although the data is not quantified, so it is impossible to know just how high your values are). Your argument that the Si signal should decrease if you are hitting other minerals besides quartz is wholly inaccurate because of the volume differences between the quartz and the micro-and nano-inclusions that it hosts. It is easy to get 100s or even 1000s of ppm of contamination when the contaminating phases have major element compositions containing things like Ca or Mg or Na etc. . ... Especially if the contaminating phases are silicates themselves, like clays or micas, as is common in low temperature hydrothermal quartz, then you would not expect Si signal to decrease when you hit lots of small inclusions. Remember, you are still ablating 98 or more % quartz. The inclusions make up a small percentage of the total volume of ablated material, but if their major element composition is dominated by things like Mg, Na, Ca, Fe, etc, then you can easily get

signals showing the presence of these elements. For some reason, you don't mention Al at all in your discussion even though you analysed it and even though many studies have shown that it is the most abundant trace element in low temperature quartz. It's not clear why Al has been ignored in favor of elements that are commonly reported as contamination in quartz from other localities. It is also perplexing why there is no CL image to accompany your LAICPMS data? A premise of your study really focuses on combining these techniques but in the end, your LACIPMS data cannot be compared directly to your CL textures or to your Raman data. I would suggest doing spot LAICPMS analyses in different regions of quartz that have been previously imaged by CL so you can compare the compositions of the red feathery quartz and the blue euhedral quartz directly. I'd expect some difference in composition, but we can't evaluate that with the given data. Only one LAICPMS line in itself is not that convincing. More LAICPMS data combined with CL imaging and preferably quantified through the use of external standards is needed to really have any confidence in the data presented. As for the Raman data, maybe I'm confused, but it doesn't seem to prove anything. As you say, you find the 503-509 peak both in feathery quartz and in euhedral quartz, so to me that means that this peak isn't necessarily telling you anything about the quartz structure in your sample. Why do you see the shoulder in euhedral quartz but then still relate it to moganite? It seems to me that if the peak is present in both euhedral quartz and moganite, then it is really of no use. Right? What am I missing? You say you did 40 analyses in your sample. Great, then a graph compiling that data would be useful and might help to clarify my question abov. How many of the spots showed the weak shoulder and is there any relation between the quartz morphology and the peaks? Quantify the relationship.

Ultimately, the results are interesting as CL textures are inherently interesting and make the world a better place, but the conclusion that feathery texture, which is actually observed even without CL, results from recrystallization of a gel is not shown or even tested in this study. It may be true, but the results here don't really support or deny this possibility. It is a conclusion suggested by previous studies and this study just seems

to accept that this is a plausible case in the case of vein quartz from this vein, but I don't see any data that necessarily proves that this is the case in the samples studied here. CL textures just show textures, not process. The LA-ICP-MS data reflects the analysis of inclusions, and even if it did reflect actual quartz elemental concentrations, LA-ICP-MS data has not been shown to be a direct indicator of quartz recrystallization in the past, and that case is not made here either. Lastly, Raman may be the most useful technique presented to demonstrate quartz recrystallization, but the data here is unclear and seems to indicate the opposite-that even regular euhedral quartz can have the 503 shoulder and that the 503 shoulder is not directly related to recrystallization of quartz or feathery texture. Thanks for the interesting study and I wish the authors best of luck in the future.

Brian Rusk Bellingham, WA, USA June 15, 2016
* * *